# Construction of boron-stereogenic compounds via enantioselective Cu-catalyzed desymmetric B–H bond insertion reaction

Guan Zhang[1,5], Zhihan Zhang[2,5], Mengyuan Hou[1], Xinping Cai[1], Kai Yang[1], Peiyuan Yu [2✉] & Qiuling Song [1,2,3,4✉]

Compared with the well-developed carbon-stereogenic chemistry, the construction of boron-stereogenic compounds remains undeveloped and challenging. Herein, the previously elusive catalytic enantioselective construction of boron-stereogenic compounds has been achieved through enantioselective desymmetric B–H bond insertion reaction. The B–H bond insertion reaction of 2-arylpyridine-boranes with versatile diazo compounds under chiral copper catalyst can afford boron-stereogenic compounds with good to excellent enantioselectivity. Moreover, the synthetic utility of this reaction is demonstrated by the scalability and downstream transformations. DFT calculations provide insights into the reaction mechanism and the origin of stereoselectivity.

[1] Key Laboratory of Molecule Synthesis and Function Discovery, Fujian Province University, College of Chemistry at Fuzhou University, Fuzhou, Fujian 350108, China. [2] Department of Chemistry and Shenzhen Grubbs Institute, Guangdong Provincial Key Laboratory of Catalysis, Southern University of Science and Technology, Shenzhen, Guangdong 518055, China. [3] Institute of Next Generation Matter Transformation, College of Materials Science Engineering at Huaqiao University 668 Jimei Boulevard, Xiamen, Fujian 361021, China. [4] School of Chemistry and Chemical Engineering, Henan Normal University, Xinxiang, Henan 453007, China. [5] These authors contributed equally: Guan Zhang, Zhihan Zhang. ✉email: yupy@sustech.edu.cn; qsong@hqu.edu.cn

**B**oron is an important metalloid element in organic chemistry. Boron typically forms tricoordinate chiral organoboron compounds with carbon-stereogenic centers, which serve as important synthetic reagents to access optically active molecules[1–5]. In addition, tricoordinate organoboron compounds can also be designed as axial chiral frameworks, efforts to asymmetric catalytic preparation of chiral organoboron compounds bearing C-B bond axis have been achieved by our group[6] and Tan & Zhang's group[7], respectively (Fig. 1A). Tricoordinate organoboron compounds have an empty p-orbital that can accept electrons from Lewis bases or nucleophiles to form tetracoordinate organoboron compounds with a tetrahedral geometry[8–15]. If the four substituents of the tetracoordinate organoboron compounds are all different, the boron atom is also a stereogenic center which is similar to the carbon, phosphine[16–19], sulfur[20–22] or silicon[23,24] center. In fact, boron-stereogenic compounds can be found in natural products[25] and materials[26,27] as well (Fig. 1B). Despite the significance of boron-stereogenic compounds, enantioselective catalytic methods for the construction of such chiral compounds are elusive, so far there is only one enantioenriched protocol reported by He et al. to procure such boron-stereogenic compounds, which was achieved by desymmetrization of diacetylative tetracoordinate boron species via a CuAAC click chemistry (Fig. 1C)[28], the other known approaches to them are restricted to the chiral resolution or chiral substrate-induced processes[29–38]. Of note, due to the small size of the boron atom (compared to very known atoms which could reveal stereogenic centers, such as carbon, nitrogen, sulfur, phosphorus etc.), it will be a challenge if the enantioselective reaction site was direct on boron itself. Thus, it leaves great space and makes the development of catalytic enantioselective synthesis of boron-stereogenic compounds with diverse structures become extremely appealing yet challenging as well.

Enantioselective desymmetrization has emerged as one of the most frequently investigated strategy to access chiral molecules[39–44]. In order to construct the boron-stereogenic center directly on the boron atom, which was distinct from the known strategy, a suitable and stable tetracoordinate organoboron substrate with two identical reaction sites is the key to the success of this goal. On the other hand, the carbene-directed B–H bond insertion reaction represents an efficient approach for the construction of new C-B bonds[45–57]. The catalytic process was pioneered by Curran[48] and Zhou & Zhu[49], and the latter two achieved the first asymmetric carbene insertion into the B−H bonds of phosphine−borane adducts (M₂HP•BH₃). Most remarkably, the targeted products are stable tetracoordinate organoboranes bearing two same B-H bonds, which perfectly meet our requirement and could serve as an ideal starting material for our purpose. Grounded on the knowledge, we envision that a metal-catalyzed enantioselective desymmetric B-H bonds insertion of such tetracoordinate boranes (L•BH₂R) with diazo compounds could render the desired enantioenriched boron-stereogenic compounds (L•B*HRR') directly on boron atom. However, this chemistry would encounter some major challenges: (1) The current B-H bond insertion reaction mainly occurs in the first B-H bond of tetracoordinate borane (L•BH₃), and the second B-H bond of tetracoordinate borane (L•BH₂R) is relatively inert and rarely investigated[48]. (2) The known tetracoordinate organobranes (L•BH₂R) are linear, which has labile nature, the choice of a suitable skeleton for tetracoordinate borane substrates would be essential for the success of the hypothesis. (3) To the best of our knowledge, there are no precedent reports that the B-X (X = C, H, O, N, P) bond is directly involved in the construction of boron-stereogenic compounds. Inspired by our previous report, cyclic tetracoordinate borons with an *N*-containing ligand would provide rigid scaffold, which will enhance the feasibility for the construction of enantioenriched boron stereogenic compounds[58].

Here, we show a copper-catalyzed enantioselective construction of boron-stereogenic center direct on boron atom via desymmetric B–H bond insertion using 2-arylpyridine-borane (L•BH₂R) as substrate with carbenes, providing a straightforward and efficient route toward boron-stereogenic compounds (Fig. 1D). DFT calculations elucidate the origin of excellent enantioselectivity and diastereoselectivity.

## Results and Discussions
To explore the proposed enantioselective desymmetric B–H bond insertion reaction, initial substrate assessing was performed with substituted 2-arylpyridine-boranes (**1**) and diazo compounds (**2**) (Fig. 2). After carefully examining various reaction parameters

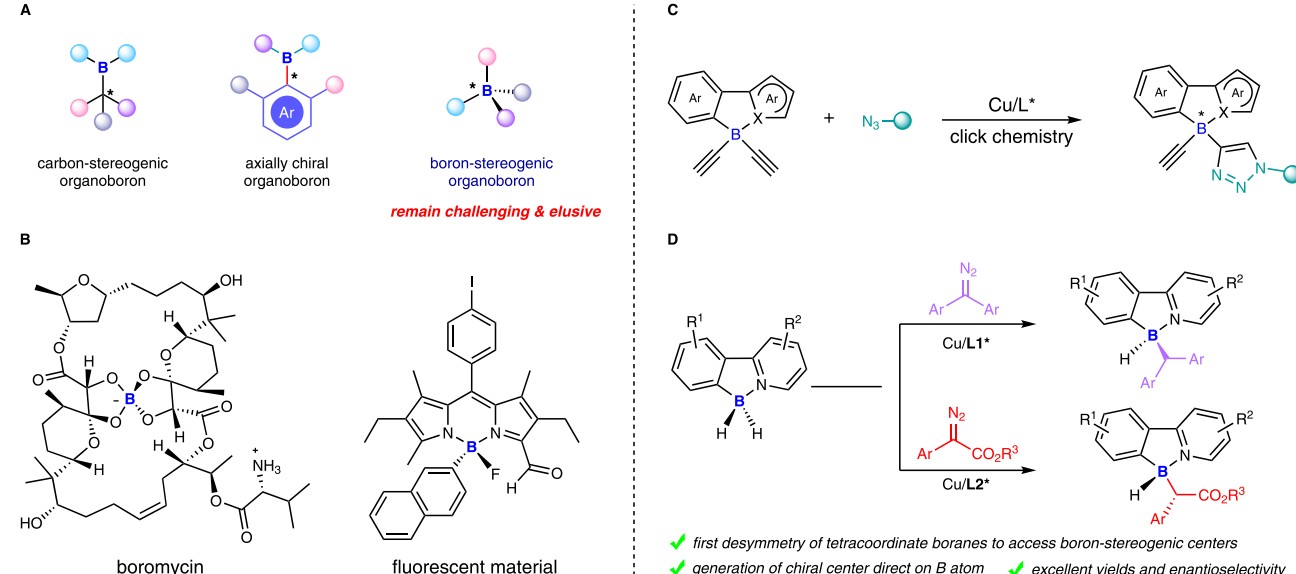

**Fig. 1 Boron-stereogenic compounds and their catalytic enantioselective synthesis. A** Current status of chiral organoboron chemistry. **B** Boron-stereogenic compounds in nature product and material. **C** Asymmetric CuAAC for the synthesis of boron-stereogenic compounds (He et al.). **D** Catalytic desymmetric B-H insertion toward boron-stereogenic compounds (this work).

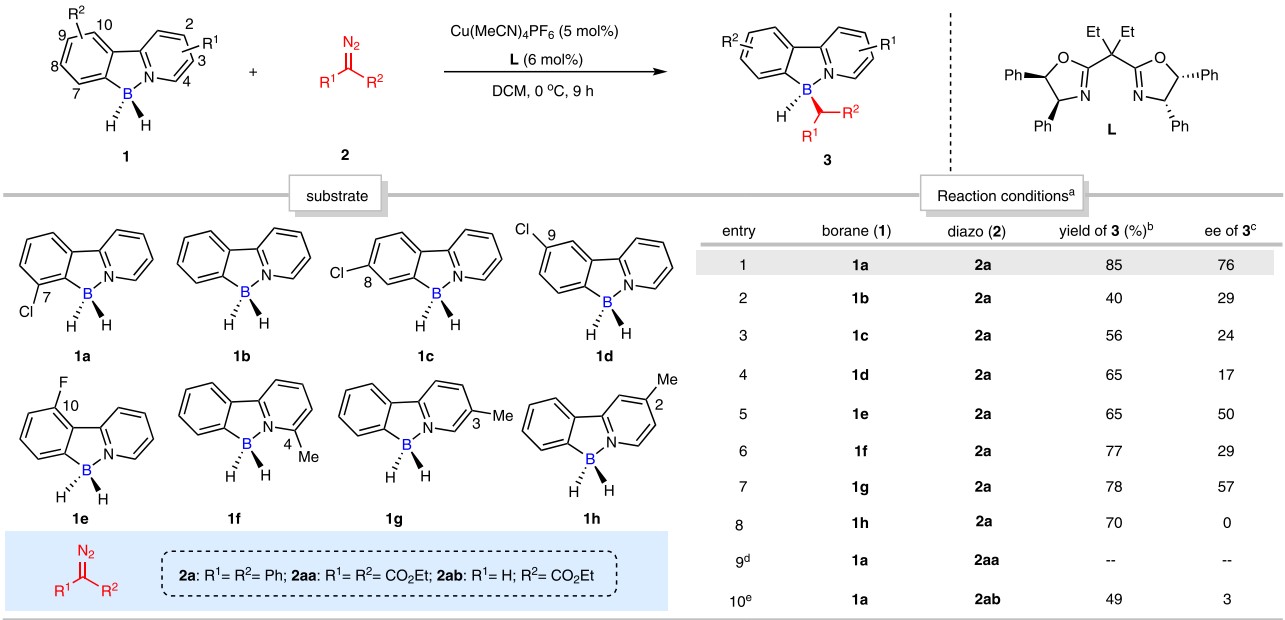

**Fig. 2 Initial substrate assessment of desymmetric B–H bond insertion reaction.** Conditions: [a] 2-arylpyridine-borane (**1**) (0.1 mmol, 1.0 equiv), diazo compound (**2**) (0.1 mmol, 1.0 equiv), Cu(MeCN)$_4$PF$_6$ (5 mol%), ligand **L** (6 mol%), solvent (1 mL) at 0 °C for 9 h. [b] Isolated yield. [c] Determined by chiral HPLC. [d] Diazo compound **2aa** instead of **2a**. [e] Diazo compound **2ab** instead of **2a**.

(entries 1-10, Fig. 2), we found that the corresponding desired boron-stereogenic product **3** was smoothly obtained using Cu(MeCN)$_4$PF$_6$ (5 mol%) as catalyst, diethyl substituted chiral bisoxazoline **L** as ligand, DCM as solvent at 0 °C under argon. Among the examined tetracoordinate boron substrates which bear substitutions on the different positions of benzene and pyridine rings (**1a-1h**), 7-chloro-substituted 2-phenylpyridine-borane (**1a**) demonstrated the best enantioselectivity with excellent yield (entry 1, Fig. 2) with (diazomethylene)dibenzene (**2a**), which suggested that the introduction of a substituent at C7-position of 2-arylpyridine-boranes (**1**) was conducive to the stereocontrol of this transformation.

In order to understand whether the benifits comes from the steric or electronic effect of the substituents on C7-position, DFT calculations were performed on this optimal result (see Supplementary Information (SI) for more details about computational methods, and Supplementary Data 1 for XYZ coordinates). Interestingly, we found that after transferring hydride to carbene center, the borane substrate was transformed to a boron cation which further underwent a barrierless S$_E$2 process delivering the final product (see Supplementary Fig. 1 in SI). The difference in barrier ($\Delta\Delta G^{\ddagger}$) with **1a** as substrate was calculated to be 1.9 kcal/mol, while $\Delta\Delta G^{\ddagger}$ with **1b** (no substituents on C7-position) was calculated to be only 0.5 kcal/mol (Fig. 3). These computational results successfully reproduced that the introduction of substituent on C7-position enhances enantioselectivity. In addition, we carried out distortion-interaction analysis[59] (see Supplementary Table 3 in SI) and found that the relative interaction energies ($\Delta\Delta E_{int}$) match well with $\Delta\Delta G^{\ddagger}$, hinting the origin of stereoselectivity and addressing the significant role of substituent on C7-position. By carefully comparing the geometries of key transition states, we found that Cl atom, which carries high electron density, points to the more congested space in TS$_{R-3a-L6}$ when borane substrate approaches catalyst, which renders longer C–H bond length with less interaction between hydride and copper-carbene (see Supplementary Fig. 2 in SI for more details). When **1b** was used as substrate, there is no essential difference in electron density between pyridine and unsubstituented phenyl ring, leading to

minor difference in barrier and poor stereoselectivity. Of note, when diazo compounds expanded to diethyl 2-diazomalonate (**2aa**) and ethyl 2-diazoacetate (**2ab**), trace or moderate amount of desired products were obtained with poor enantioselectivities (entries 9-10, Fig. 2).

With the preliminary substrate-assessment results in hand, we decided to choose 7-chloro-substituted 2-phenylpyridine-borane (**1a**) and (diazomethylene)dibenzene (**2a**) as model substrates to perform further and prudent condition evaluations. More chiral oxazolines (**L1**-**L6**) were tested and it turned out that ligand **L1** demonstrated the best reactivity and stereoselectivity (entries 1-6, Fig. 4) (see Supplementary Tables 1 and 2 in SI for more ligands). Subsequent solvent and temperature examinations indicated that DCM and -35 °C could render an optimal result with the generation of the desired product **3a** in 95% yield with 91% ee (entries 7-10, Fig. 4).

Having optimized the model desymmetric B–H bond insertion reaction with symmetric diazo compounds (Figs. 2 and 4), we first examined the substrate generality of these conditions with respect to different symmetric diazo compounds (Fig. 5). In general, a series of substituted diaryl diazomethanes **2a-2k** reacted smoothly with 7-Cl-2-phenylpyridine-boranes (**1a**), producing the corresponding enantioenriched boron-stereogenic compounds **3a-3k** in moderate to excellent yields and good to excellent enantioselectivities. The halogens (**3b-3d**, **3 h**), methoxy group (**3e-3f**), methyl group (**3 g**) as well as symmetric poly-phenyl cyclic rings (**3h-3k**) all participated efficiently in this process, yielding the target boron-stereogenic molecules in good to excellent enantioselectivities (83-98% ee). Next, a series of 2-arylpyridine-borane derivatives (**3l-3s**) were tested to investigate the substrate scope of this enantioselective B–H bond insertion reaction with (diazomethylene)dibenzene (**2a**). It is of note that 2-arylpyridine-boranes bearing various halogens all proceeded smoothly under the standard conditions with good to excellent enantioselectivities (**3l-3r**, 88-92% ee). Disubstituted 2-arylpyridine-boranes could furnish enantioenriched boron-stereo-genic compounds (**3n-3r**) with high yields (79-95%) and enantioselectivities (88-91% ee). 2-Naphthylpyridine-borane (**1s**) also provided the corresponding product **3s** in 90% yield with 93% ee. The structure

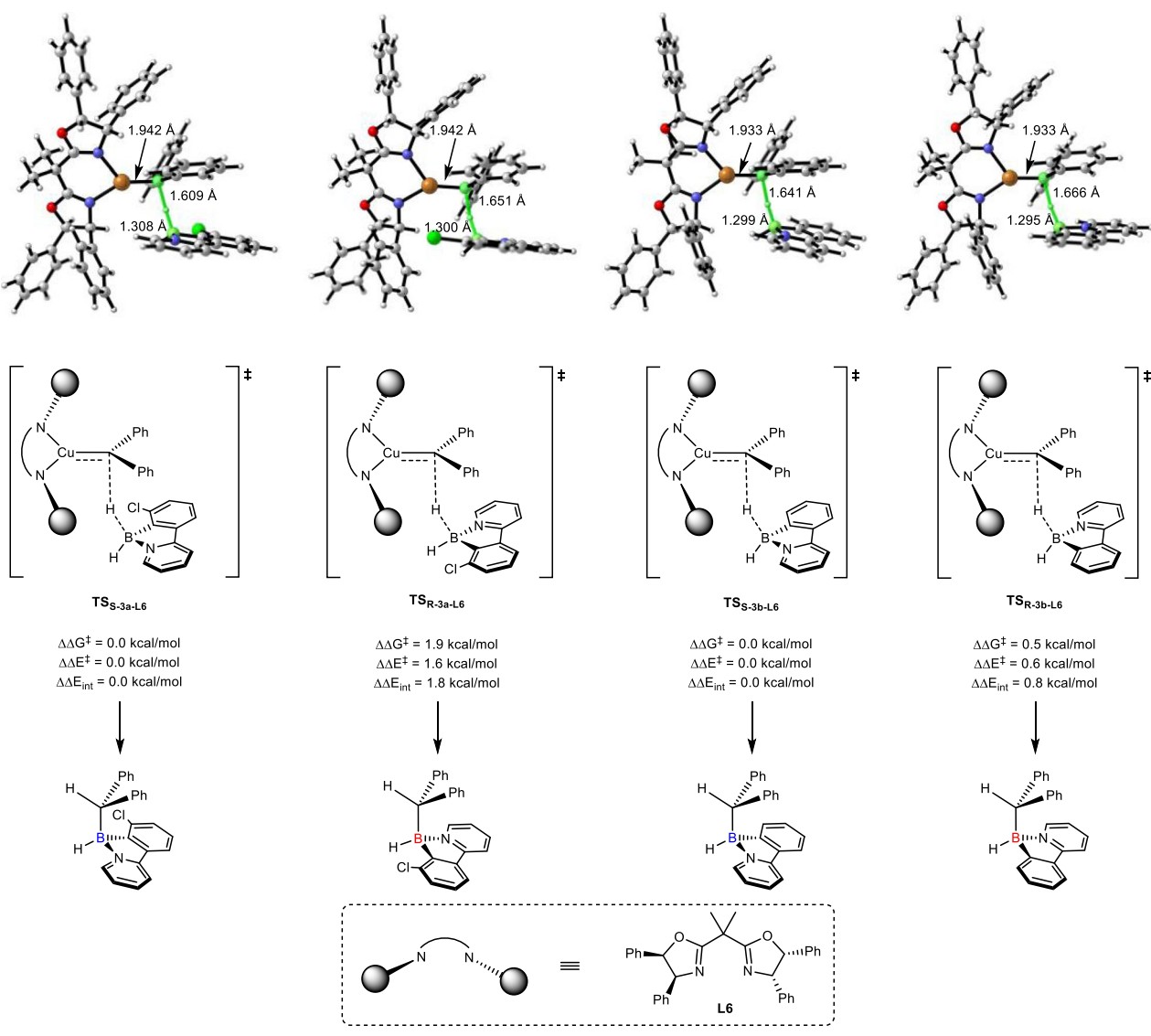

**Fig. 3 DFT-calculated key transition state structures for the enantio-determining hydride transfer step with 1a and 1b as substrates.** All computational results are performed at the M06/SMD/def2TZVP//B3LYP/SMD/def2SVP level of theory.

and absolute configuration of compound **3n** was determined by X-ray crystallographic analysis (CCDC 2125214).

Encouraged by the outcomes of this enantioselective desymmetric B-H bond insertion reaction with symmetric diaryl diazomethanes (Fig. 5), which only generated one B-stereogenic center in the final products, we prompted to explore the feasibility of 7-chloro-substituted 2-phenylpyridine-borane (**1a**) with unsymmetric diazo compound, specifically, ethyl α-diazoarylacetates **4a**, since two consecutive stereogenic centers would be resulted and in asymmetric synthesis, such protocols are always a big challenge with both high enantioselectivity and diastereoselectivity. To our delight, with the rapid evaluation of chiral bisoxazolines, ligand **L2** demonstrated the best enantioselectivity (98% ee) and diastereoselectivity (dr > 20:1) (entries 1-6, Fig. 6). Further solvent screening indicated that the best reaction efficiency was endowed by DCE (entries 6-8, Fig. 6). It should be noted that this reaction was diastereoselective even in the absence of chiral ligand (entry 9, Fig. 6), however, the ratio of the two diastereomers are only 2:1, with the help of chiral ligand, both enantioselectivity and diastereoselectivity are dramatically improved (entry 2, Fig. 6).

Interestingly, the diastereomers of this reaction were isolable. And the absolute configuration of **5a** was unambiguously assigned by X-ray crystallographic analysis (CCDC 2104867).

Having arrived at the optimal conditions with unsymmetric ethyl α-diazoarylacetate **4a** (entry 7, Fig. 6), we first examined the substrate generality of these conditions with respect to different 2-arylpyridine-boranes **1** with ethyl α-diazoarylacetate (**4a**) (Fig. 7). DFT calculations suggested that substituents on C7 position of 2-arylpyridine-boranes are very important in order to construct enantioenriched tetracoordinate boron species. Like the aforementioned symmetric carbene insertion, the origin of the chirality of B-stereogenic center is resulted from the interaction between chiral environment of copper-carbene and the asymmetric steric structure of borane, which is not sorely induced by the construction of C-stereogenic center. Gratifyingly, our experimental outcomes perfectly validated the DFT calculations. For instance, compared to the chiral product **5b** which was generated with 7-fluoro-2-phenylpyridine-borane, 8-fluoro-2-phenylpyridine-borane delivered the corresponding product **5c** with good enantioselectivity (90% ee) while much lower dr value (>20:1 vs 3.3:1). In general, a series of 7-substituted 2-phenylpyridine-boranes **1**

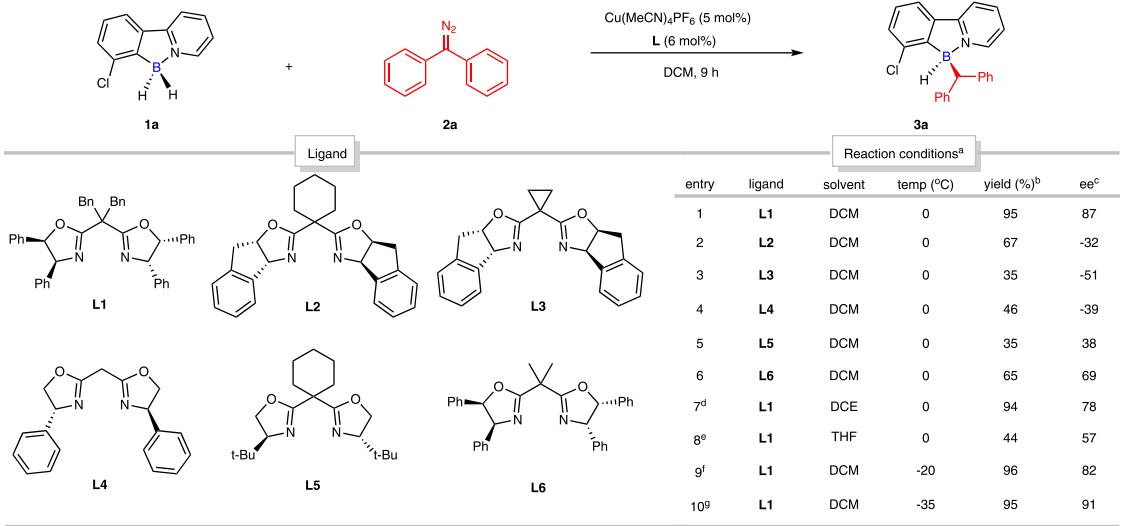

| entry | ligand | solvent | temp (°C) | yield (%)[b] | ee[c] |
|-------|--------|---------|-----------|-----------|-------|
| 1 | L1 | DCM | 0 | 95 | 87 |
| 2 | L2 | DCM | 0 | 67 | -32 |
| 3 | L3 | DCM | 0 | 35 | -51 |
| 4 | L4 | DCM | 0 | 46 | -39 |
| 5 | L5 | DCM | 0 | 35 | 38 |
| 6 | L6 | DCM | 0 | 65 | 69 |
| 7[d] | L1 | DCE | 0 | 94 | 78 |
| 8[e] | L1 | THF | 0 | 44 | 57 |
| 9[f] | L1 | DCM | -20 | 96 | 82 |
| 10[g] | L1 | DCM | -35 | 95 | 91 |

**Fig. 4 Optimization of desymmetric B–H bond insertion reaction with (diazomethylene)dibenzene (2a).** Conditions: [a] 7-Cl 2-phenylpyridine-borane (**1a**) (0.1 mmol, 1.0 equiv), (diazomethylene)dibenzene (**2a**) (0.12 mmol, 1.2 equiv), Cu(MeCN)$_4$PF$_6$ (5 mol%), ligand (6 mol%), solvent (1 mL) at 0 °C for 20 h. [b] Isolated yield. [c] Determined by chiral HPLC. [d] DCE instead of DCM. [e] THF instead of DCM. [f] -20 °C instead of 0 °C. [g] -35 °C instead of 0 °C.

**3a**, 95%, 91% ee

**3b**, 93%, 91% ee

**3c**, 96%, 92% ee

**3d**, 98%, 89% ee

**3e**, 88%, 86% ee

**3f**, 97%, 90% ee

**3g**, 95%, 86% ee

**3h**, 90%, 83% ee

**3i**, 87%, 98% ee

**3j**, 57%, 93% ee

**3k**, 59%, 91% ee

**3l**, 85%, 92% ee

**3m**, 93%, 89% ee

**3n**, 79%, 90% ee

X-ray of **3n**

**3o**, 95%, 89% ee

**3p**, 82%, 91% ee

**3q**, 90%, 88% ee

**3r**, 95%, 91% ee

**3s**, 90%, 93% ee

**Fig. 5 Scope of enantioselective B−H insertion with symmetric diazo compounds.** Conditions: [a] 2-arylpyridine-borane **1** (0.1 mmol, 1.0 equiv), symmetric diaryl diazomethane **2** (0.12 mmol, 1.2 equiv), Cu(MeCN)$_4$PF$_6$ (5 mol%), **L1** (6 mol%), DCM (1 mL) at -35 °C for 9 h. [b] Isolated yield.

**Fig. 6 Optimization of desymmetric B–H bond insertion reaction with ethyl α-diazophenylacetate (4a).** Conditions: [a] 2-arylpyridine-borane (**1a**) (0.12 mmol, 1.2 equiv), ethyl α-diazophenylacetate (**4a**) (0.1 mmol, 1 equiv), Cu(MeCN)$_4$PF$_6$ (5 mol%), ligand (6 mol%), solvent (1 mL) at 0 °C for 20 h. [b] Isolated yield. [c] Determined by [1]H NMR. [d] Determined by chiral HPLC.

reacted smoothly with ethyl α-diazophenylacetate (**4a**), the corresponding enantioenriched boron-stereogenic compounds **5b**, **5d**-**5l** were procured in good to excellent yields and enantioselectivities as well as excellent diastereoselectivities (dr > 20:1). For example, a variety of halogen substituents were tolerated well, all yielding the target boron-stereogenic molecules **5b**, **5d**-**5k** in both excellent enantioselectivities (93-99% ee) and diastereoselectivities (>20:1 dr), no matter they were mono- or disubstituted ones. The methyl (**5 g**, **5k** and **5 m**) and trifluoromethoxy (**5 l**) groups substituted 2-phenylpyridine-boranes all participated efficiently in this process, only compound **5 m** rendered with a relative poor dr value (6.5:1). To our delight, 2-heteroarylpyridine-boranes were also good substrates for this enantioselective desymmetric B–H bond insertion reaction, boron-stereogenic products **5n** and **5o** were obtained in good yields with good diastereoselectivities and excellent enantioselectivities. The introduction of substituent at the 1 position of pyridine ring had minimal influence on the reactivity and enantioselectivity of this transformation, however, it lowered the diastereoselectivity (**5p**, 91% ee vs 3:1 dr). Next, the scope of α-diazoarylacetates **4** was investigated. Changing the ester moiety of α-diazoarylacetates to methyl, benzyl, *tert*-butyl and phenyl did not change the excellent outcomes, they were all good candidates to deliver the corresponding product **5q**-**5t** in excellent enantioselectivities and diastereoselectivities. Both electron-rich (**5u**-**5y**) and electron-poor (**5z**, **5aa**) groups, as well as halogens (**5ab**-**5ad**) on the benzene ring of α-diazoarylacetates underwent this enantioselective desymmetric B–H bond insertion reaction with moderate to excellent enantioselectivities and excellent diastereoselectivites. Moreover, α-diazo 2-naphthylacetates (**5ae**) was also compatible to this reaction.

**Synthetic application**. To demonstrate the synthetic utility of this enantioselective desymmetric B–H bond insertion reaction (Fig. 8), we first performed a gram-scale reaction. To our delight, this reaction shows similar efficiency and enantioselectivity under the condition of halving the catalyst (Fig. 8A), which indicates that the large-scale chemical production of the boron-stereogenic compound could be achieved. In addition, the derivatizations of boron-stereogenic compound **5a** were also conducted to further

showcase the synthetic utility. Gratifyingly, the boron-stereogenic compound **5a** could further undergo B–H bond insertion reaction, affording 3-carbon substituted boron-stereogenic compound **6** with the retention of chirality (98% ee). Moreover, the ester moiety of **5a** could also be reduced to the alcohol **7** in the presence of DIBAl-H. It should be noted that the bromide **5d** could be easily converted to complex **8** by palladium-catalyzed cross-coupling (Fig. 8B). Remarkably, we could combine the double B–H bond insertion reactions with a two-step in one-pot protocol (Fig. 8C). Without further purification after the enantioselective desymmetric B–H bond insertion between **1a** and **4a**, the second diazo compound **9** was directly added to the reaction mixture in the absence of new catalyst and new ligand, the desired product **6** was obtained in 76% isolated yield with 97% ee.

**Mechanistic studies**. To explore the reaction mechanism, some isotope experiments were carried out (Fig. 9). When deuterated 2-phenylpyridine-borane **9-d₂** was used as the substrate, the deuterium atom was transferred to the α-carbon of the boron-stereogenic product **9-d₂** (Fig. 9A). Moreover, the kinetic isotope effect (KIE) experiment ($k_H/k_D = 1.5$) indicated that B–H bond insertion process (including B–H bond cleavage and C–H bond formation) might not be involved in a rate-determining step (Fig. 9B), and this result was consistent with previous report[52].

DFT calculations (Fig. 10) show that when dichloro-substituented substrate **1n** was used, **TS$_{SR-5h-L3}$**, the transition state ultimately leading to the experimentally observed product **5 h**, is the most favorable one among four different configurations. Distortion-interaction analysis[59] (see Supplementary Table 4) reveals that the relative distortion energies of the copper-carbene ($\Delta\Delta E_{dist(carbene)}$) are the dominant contributor to the difference in relative barriers of reaction ($\Delta\Delta G^{\ddagger}$). The orientation of the ester group in **TS$_{SR-5h-L3}$** is preserved compared to the optimized structure of copper carbene. In contrast, the ester group in **TS$_{SS-5h-L3}$** rotates away from ideal geometry by around 10° to avoid strong repulsion of Cl atom, causing an elevation of Gibbs energy (see Supplementary Fig. 3). Similarly, copper-carbene fragment in **TS$_{RS-5h-L3}$** contains more congested space, rendering itself more sensitive to the entrance of borane substrate (see Supplementary Fig. 4). A larger distortion of

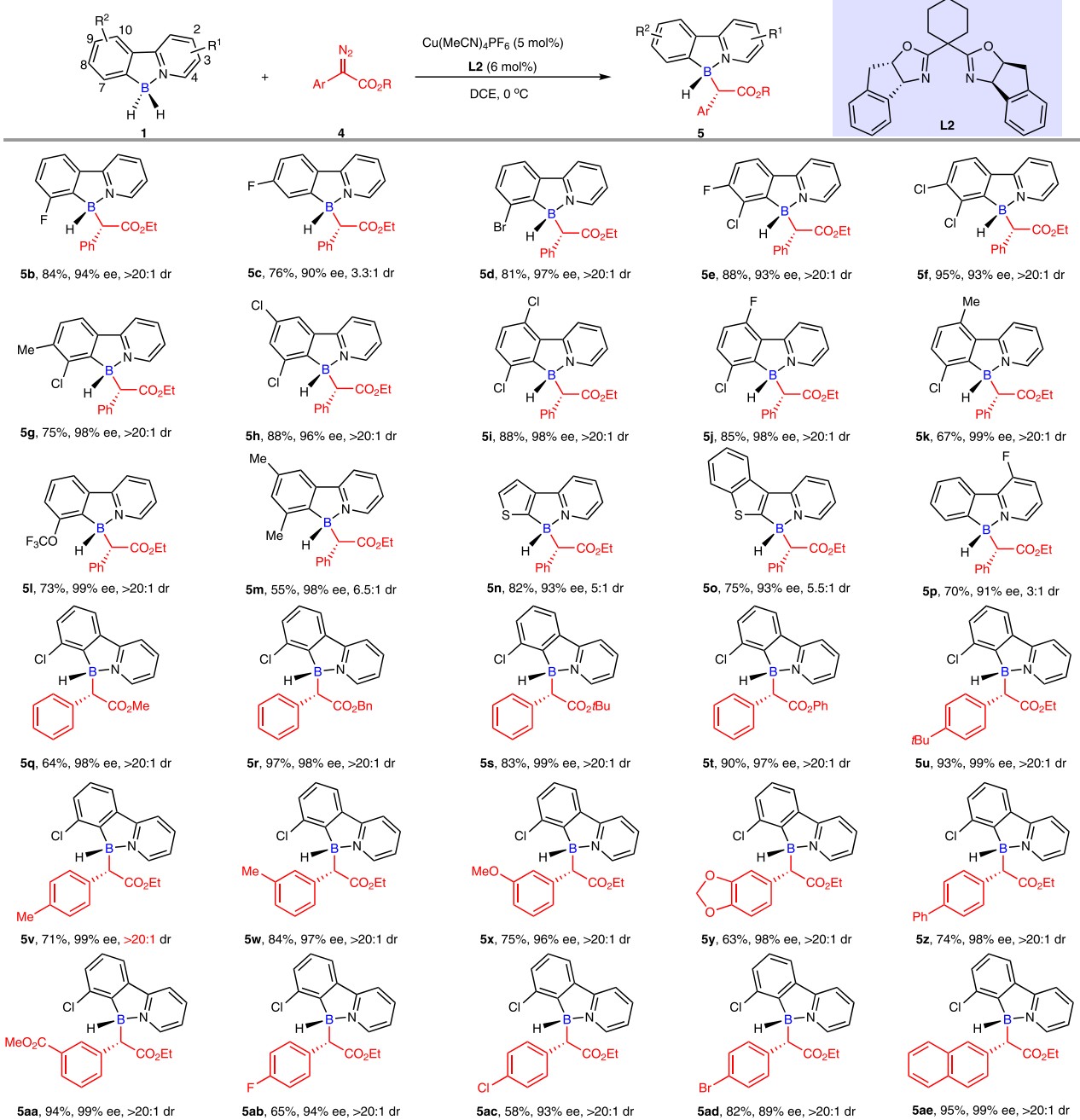

**Fig. 7 Substrate scope of 2-arylpyridine-boranes with α-diazoarylacetates.** Conditions: [a] 2-arylpyridine-borane (**1**) (0.12 mmol, 1.2 equiv), α-diazoarylacetate **4** (0.1 mmol, 1 equiv), Cu(MeCN)$_4$BF$_4$ (5 mol%), **L2** (6 mol%), DCE (1 mL) at 0 °C. [b] Isolated yield.

ligand in **TS$_{RS-5h-L3}$** could explain the larger distortion energy of the copper-carbene fragment and the corresponding difference in the barrier of reaction. When **1b** was used as substrate, although enantioselectivity is still high due to the similar reason for **1n** (Supplementary Fig. 5), the diastereomeric ratio drops from more than 20:1 to 3.3:1. This interesting result is consistent with the decrease in the calculated $\Delta\Delta G^{\ddagger}$ and the relative distortion energies of the copper-carbene ($\Delta\Delta E_{dist(carbene)}$). The orientation of the ester group are similar for **TS$_{SR-11-L3}$** and **TS$_{SS-11-L3}$**, since it is hard for copper carbene to differentiate unsubstituented phenyl ring and pyridine in the absence of Cl atom on C7-position (Supplementary Fig. 6). Both experimental and computational results articulates the vital role of C7-substitution for improving diastereoselectivity.

In conclusion, we have achieved a catalytic protocol for the construction of boron-stereogenic compounds, via enantioselective copper-catalyzed desymmetric B–H bond insertion reaction of 2-arylpyridine-boranes with diazo compounds, and a series of enantioriched boron-stereogenic compounds were efficiently synthesized with high enantioselectivity and diastereoselectivity. Both experiments and DFT calculations suggested that the introduction of a substituent at C7-position of 2-arylpyridine-boranes was conducive to the stereocontrol of this transformation. Moreover, 2-arylpyridine-borane could also undergo twice B-H bond insertion reactions to obtain 3-carbon substituted boron-stereogenic compound. Further investigations to the utility of these boron-stereogenic molecules are currently underway in our laboratory.

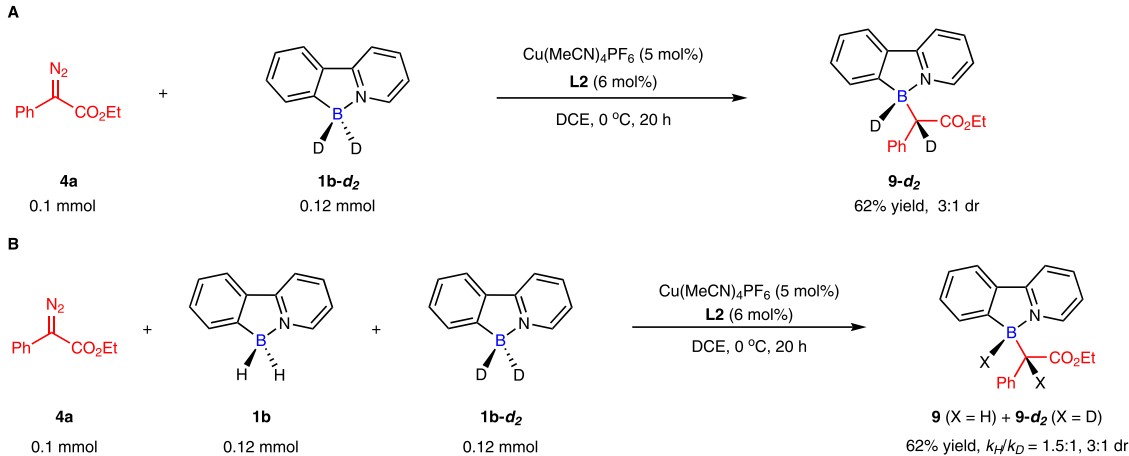

**Fig. 8 Gram-scale reaction and transformations. A** Gram-scale reaction. **B** Transformations. **C** One-pot reaction to access 3-carbon substituted boron-stereogenic compound **6**.

**Fig. 9 Isotope labeling experiments. A** Deuterium labeling experiment. **B** KIE experiment.

## Methods

**General procedure A for the boron-stereogenic compounds with 2-arylpyridine-borane and diaryl diazomethane**. In air, a 10 mL schlenk tube was charged with **1** (0.10 mmol, 1.0 equiv), **2** (0.12 mmol, 1.2 equiv), Cu(MeCN)$_4$PF$_6$ (5 mol%), **L1** (6 mol%). The tube was evacuated and filled with argon for three cycles. Then, 1 mL of DCM was added under argon. The reaction was allowed to stir at -35 °C for 9 hours. Upon completion, proper amount of silica gel was added to the reaction mixture. After removal of the solvent, the crude reaction mixture was purified on silica gel (petroleum ether and ethyl acetate) to afford the desired products.

**General procedure B for the boron-stereogenic compounds with 2-arylpyridine-borane and α-diazoarylacetate**. In air, a 10 mL schlenk tube was charged with **1** (0.12 mmol, 1.2 equiv), **4** (0.1 mmol, 1 equiv), Cu(MeCN)$_4$PF$_6$ (5 mol%), **L2** (6 mol%). The tube was evacuated and filled with argon for three cycles. Then, 1 mL of DCE was added under argon. The reaction was allowed to stir at 0 °C for 20 hours. Upon completion, the proper amount of silica gel was added to the reaction mixture. After removal of the solvent, the crude reaction mixture was purified on silica gel (petroleum ether and ethyl acetate) to afford the desired products.

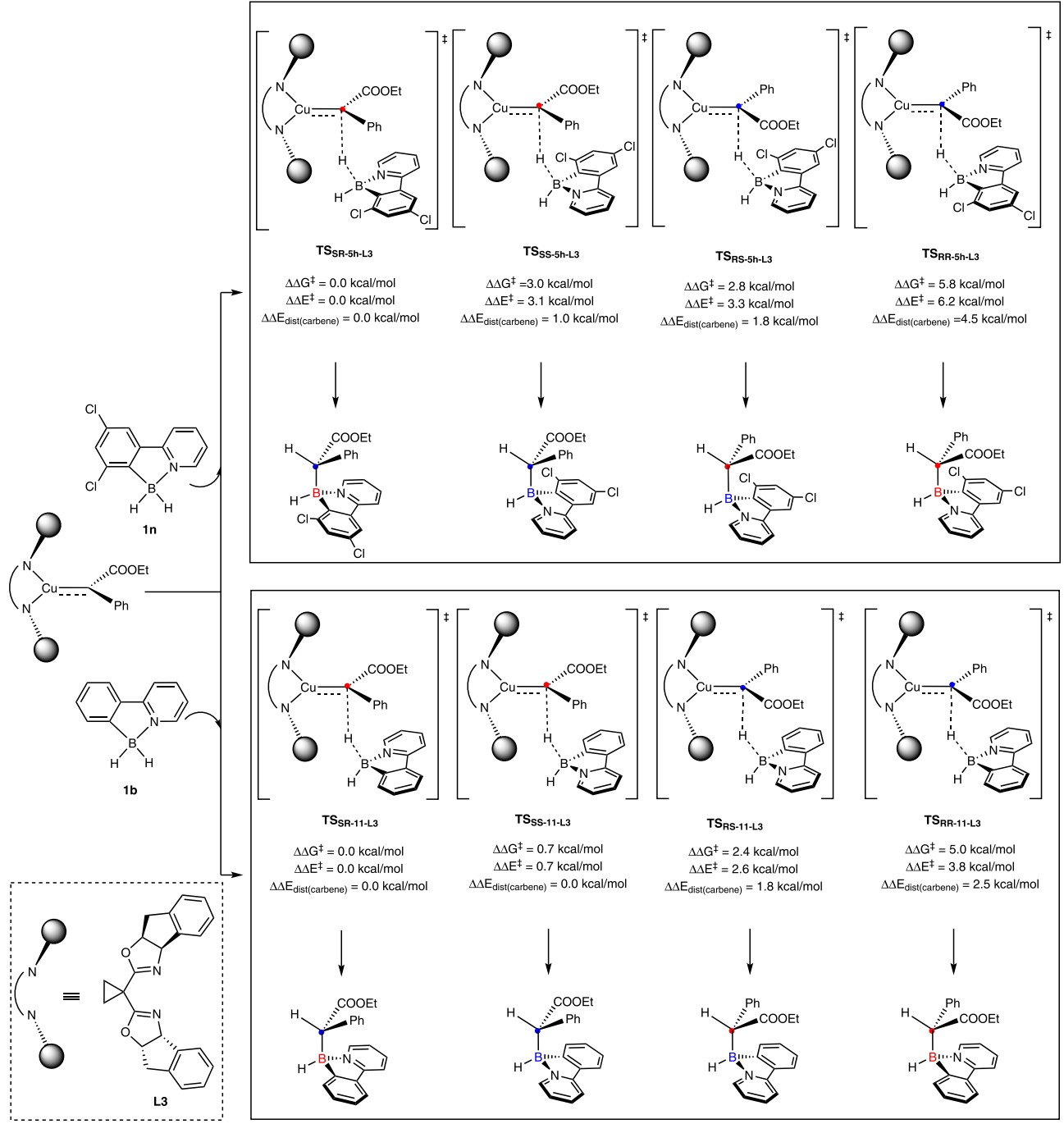

**Fig. 10 Computational investigation on the construction of two consecutive chiral centers.** All computational results are performed at the M06/SMD/def2TZVP//B3LYP/SMD/def2SVP level of theory.

## Data availability

The X-ray crystallographic coordinates for structures of **3n** and **5a** reported in this Article have been deposited at the Cambridge Crystallographic Data Centre (CCDC), under deposition numbers CCDC 2125214 (**3n**) and 2104867 (**5a**). These data can be obtained free of charge from http://www.ccdc.cam.ac.uk/data_request/cif (The Cambridge Crystallographic Data Centre). Experimental procedures, characterization of new compounds, and DFT calculations (see Supplementary Data 1 for XYZ coordiantes) are available in the Supplementary information.

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

## Acknowledgements

Financial support from National Natural Science Foundation of China (21931013) (to Q.S.), Guangdong Provincial Key Laboratory of Catalysis (No. 2020B121201002) (to Q.S. and P.Y.) and Open Research Fund of School of Chemistry and Chemical Engineering, Henan Normal University (to Q.S.) is gratefully acknowledged. Computational work was supported by Center for Computational Science and Engineering at Southern University of Science and Technology, and the CHEM high-performance supercomputer cluster (CHEM-HPC) located at the Department of Chemistry, Southern University of Science and Technology.

## Author contributions

Q.S. conceived and directed the project. G.Z. & M.H. performed experiments and prepared the supplementary information. X.C. & K.Y. helped collecting some new compounds analyzing the data. P.Y. and Z.Z. performed the DFT calculations and drafted the DFT parts. Q.S., P.Y., G.Z. & Z.Z. wrote the paper. All authors discussed the results and commented on the manuscript.

## Competing interests

The authors declare no competing interests.
