## [Peer Review File · Nature Communications]

REVIEWER COMMENTS

Reviewer #1 (Remarks to the Author):

In this work, Song and co-workers developed the first asymmetric construction of boron-stereogenic center direct on boron atom via [Cu]-catalyzed enantioselective desymmetric B–H bond insertion reaction. By utilizing well designed 2-arylpyridine-boranes as substrates, this desymmetric reaction proceeded smoothly with symmetric and unsymmetric diazo compounds, affording the corresponding chiral boron-stereogenic compounds with high enantioselectivity and diastereoselectivity. In addition, the isotope labeling experiments and DFT calculations were well performed, which greatly contributed to the elucidation of the reaction mechanism. Moreover, the derivatizations of these boron-stereogenic molecules were also carried out, in which a secondary B–H bond insertion reaction is a highlight, demonstrating the high reactivity of this class of skeletons. Therefore, I think this well-written manuscript is suitable for publication in Nat. Commun. as a research article after minor revisions as described below:

1. In Table 2, as well as in Table 3, please double check whether the generated products were all 3a or 5a, because the reactions with ligands L1 and L6 were very likely to produce products with opposite configurations.
2. The reaction between substrates 1p and 2k should be performed to see if the obtained product has the axial chirality.
3. The direction of hashed wedged bond is incorrect in some compounds (5u, 5w, 5y,...).
4. Compared with Scheme 1, the yields of Scheme 2 are generally lower. Can the byproduct with successive B–H insertion be observed? That may be caused by the relatively high activity of donor-acceptor carbenes (similar with 5a to 6 in Scheme 3B).
5. In the Experimental Section, ¹⁹F NMR data should be included for all F-substituted compounds, and the corresponding spectra should also be included.

Reviewer #2 (Remarks to the Author):

In this manuscript, the authors describe an enantioselective synthesis of chiral-at-boron compounds via Cu-catalyzed carbene insertion reactions with tetracoordinated borane-pyridine substrates by

using a desymmetrization strategy. This reviewer agrees that although the enantioselective carbene insertion to B–H bonds has recently been studied extensively, the synthesis of chiral-at-born compounds via this method remained elusive. The reaction design of this work is quite smart in that the utilization of rigid tetracoordinated borane-pyridines as substrates should enhance the chiral discrimination during the key B–H insertion step. Thus, this reviewer recognizes the novelty of this manuscript for published in Nat. Commun. However, a few minor revisions should be addressed before the final acceptance.

1. The authors emphasized the importance of the C7-substitution on the borane-pyridine substrates in asymmetric induction. A plausible model was also proposed by performing DFT calculations using both C7-substituted and C7-unsubstituted substrates. However, it should be noted that thiophene- and benzothiophene-derived substrates also participated in the reaction, leading to good enantioselectivity (5n, 5o, 93% ee). The structures of these substrates are quite different from that of the model substrate. Thus, more comments on this phenomenon should be expected in the revised manuscript.

2. This reviewer does not understand on what basis the authors can draw the absolute configuration of the minor product 5a' explicitly in Table 3. Although DFT calculations showed the energy order of the transition states leading to the two enantiomers (TSSS-5h-L3 and TSRR-5h-L3, Figure 3), this cannot be used as the sole decisive evidence for the determination of absolute configuration. This reviewer suggests not to draw the absolute configuration of 5a' if SC-XRD data are not available.

3. The computational level of theory should be described in the main text and the captions of Figures 2 and 3.

4. Further language polishing is necessary since the text is not fluent and grammatical inaccuracy is often encountered: "...but there are also shadows of boron-stereogenic compounds in chiral catalysts and materials" (Page 2); "Motivation and Construction of Boron-stereogenic Compounds Direct on Boron Atom." (Caption of Figure 1) ...

5. Author contributions: "Q.S. & Z.F. wrote the paper". However, these is not a coauthor whose name can be abbreviated as Z.F.

Reviewer #3 (Remarks to the Author):

Dear Editor,

Yu, Song and co-workers described in this article the "Construction of boron stereogenic compounds via enantioselective Cu-catalyzed desymmetric B-H bond insertion reaction".

This paper describes the construction of compounds with a stereogenic boron atom via a copper-catalyzed enantioselective method, resulting from the insertion of diazo compounds into 2-arylpyridine boranes. The method seems general, and the synthesized derivatives are obtained in good yields and with very good enantioselectivities. The use of prochiral diazo reagents also allows to obtain compounds with contiguous stereogenic boron and carbon atoms with very good enantio- and diastereo-selectivities. This study is completed by experimental and theoretical mechanistic studies on the origin of the selectivity, corroborating the experimental results.

Importantly, this article is quite original. It is the second method known in the literature to obtain a stereogenic boron atom by an enantioselective route. Although the reaction of insertion of diazo compounds into the B-H bond is a reactivity that has been described previously, notably by Curran and others, the use of this method here is quite original and leads to a very high control of the enantioselectivity/diastereoselectivity. The results are more or less well presented, the experimental section is complete and well described, and data as NMR spectral data or HPLC traces are of good quality.

However, there are some points that definitely need to be improved.

1- As a major point, the manuscript should be proofread by an English speaker. Overall, the sentences are too long, and sometimes become difficult to read and understand, and the manuscript suffers from this.

2- The term asymmetric should be removed from many sentences, as it is redundant with good stereodescriptive terms such as enantio- or diastereoselective:

*in the abstract: "The previously elusive asymmetric catalytic construction of boron-stereogenic compounds has been achieved through enantioselective desymmetric B-H bond insertion reaction herein"

*in the manuscript: for example, P4, replace asymmetric by enantioselective or in conclusion (P15) remove asymmetric, etc...

3- The authors might need to soften some of their claims concerning:

"Of note, the reported asymmetric catalytic strategy achieved by a CuAAC click chemistry by desymmetrization of diacetylative tetracoordinate boron species, it did not directly generate a boron-stereogenic center on the boron atom"

If the authors refer to He's paper (JACS 2021, 143, 16302), this statement is false. The "click" reaction described by He et al. is indeed enantioselective, the boron atom becomes stereogenic in this step. This statement should be deleted or rewritten as it is awkward. Perhaps they meant that they did not directly substitute one group with another directly on the boron atom to reveal a stereogenic boron center?

4- Comments on Figure S1, wrongly stated: "IRC calculations verified that the transition state links reactant and product and after hydride transfer step, the boron cation attack carbon to deliver product without any energy penalty". It is the carbon atom which is nucleophilic (or the organometallic copper species) and which adds itself to the borenium ion, and not the reverse.

5- One point needs to be clarified concerning the origin of diastereoselectivity. The sentence “There is a synergistic effect between the two stereogenic centers (B and C atoms) instead of inducing effect originated from C-stereogenic center to consecutive B-stereogenic center by the enantioselective B–H bond insertion reaction” is not clear. It should be explained and detailed.

6- In the ESI, HPLC traces of the racemic compounds from Table 3 and Scheme 2 show only one pair of enantiomers of the major diastereomer. However, without chiral ligand, the d.r. is 2:1 (Table 3). Where is the trace of the second diastereomer?

If the racemic version of the reaction is diastereoselective, this calls the proposed mechanism into question. This point should be reviewed, and the HPLC chromatograms should be shown in full if this was not the case.

6- It misses diastereomeric ratio in schemes 3 and 4

After major revisions, I think that the manuscript has the quality, interest, and novelty to be published in Nature Communications.

Reviewer #4 (Remarks to the Author):

Song et al. report an enantioselective copper-catalyzed desymmetric B–H bond insertion reaction of 2-arylpyridine-boranes with diazo compounds to construct boron-stereogenic compounds and the stereocontrol of this transformation was suggested by experiments and DFT calculations. However, this strategy is just a derivation of He’s work (J. Am. Chem. Soc. 2021, 143, 16302). They both employed the same catalyst and similar substrates and ligands. The main highlight of this manuscript is switching to B–H bond insertion from CuAAC reaction but both diazo compounds and azides are relative hazardous. The substrate scope is quite narrow. Relatively high enantioselectivity and diastereoselectivity are limited to substrates substituted on C7 which is similar to the results of He’s work too. In the part of synthetic utility demonstration, only double B–H bond insertion, reduction of ester and Suzuki coupling reaction are discussed. The author should make comments on the real usefulness of the product therein formed in detail.

In consideration of these aspects, the referee recommends the manuscript should be published in a more specialized journal.

Point-by-Point Response

Reviewers' Comments:

Reviewer #1 (Remarks to the Author):

In this work, Song and co-workers developed the first asymmetric construction of boron-stereogenic center direct on boron atom via [Cu]-catalyzed enantioselective desymmetric B–H bond insertion reaction. By utilizing well designed 2-arylpyridine-boranes as substrates, this desymmetric reaction proceeded smoothly with symmetric and unsymmetric diazo compounds, affording the corresponding chiral boron-stereogenic compounds with high enantioselectivity and diastereoselectivity. In addition, the isotope labeling experiments and DFT calculations were well performed, which greatly contributed to the elucidation of the reaction mechanism. Moreover, the derivatizations of these boron-stereogenic molecules were also carried out, in which a secondary B-H bond insertion reaction is a highlight, demonstrating the high reactivity of this class of skeletons. Therefore, I think this well-written manuscript is suitable for publication in Nat. Commun. as a research article after minor revisions

as described below:

Response: We sincerely thank this reviewer for the favorable comments on our work, we really appreciate it.

1. In Table 2, as well as in Table 3, please double check whether the generated products were all **3a** or **5a**, because the reactions with ligands L1 and L6 were very likely to produce products with opposite configurations.

Response: We deeply thank this reviewer for careful readings and pointing it out, in Table 2, opposite configurations of the product **3a** were obtained with ligands L2, L3 and L4, configuration of ligand L6 was incorrect, it has been revised. In Table 3, opposite configurations of the product **5a** were obtained with ligands L1, L5 and L6, again, configuration of ligand L6 was incorrect, it has been revised. All of these problems have been corrected, please see our revised manuscript.

2. The reaction between substrates **1p** and **2k** should be performed to see if the obtained product has the axial chirality.

Response: We sincerely thank this reviewer for pointing it out. Per the request, the reaction between substrates **1p** and **2k** were performed and the corresponding product could not be obtained, since large amounts of starting materials remained (see below), therefore we could not determine whether the product has axial chirality.

However, based on our knowledge about axial chirality, the current most common ones and prevalent types of axial chirality exist between two sp^2 hybridized atoms, such as Csp^2-Csp^2 , Csp^2-Nsp^2 , Csp^2-Bsp^2 and Nsp^2-Nsp^2 bonds. Very recently, one example of axial chirality was reported between Csp^2-Csp^3 bonds, yet the structure was very unique (*Nat. Catal.* **2021**, *4*, 457-462). In our product, the axis is between two sp^3 hybridized atoms (one is sp^3 hybridized boron and another one is sp^3 hybridized carbon), and both of the atoms have one H atom attached on them, there is no precedent about the axial chirality like this case between two sp^3 hybridized atoms. Therefore, it is really difficult to identify whether the product have axial chirality. In our cases, we focus on the boron-stereogenic center.

3. The direction of hashed wedged bond is incorrect in some compounds (**5u**, **5w**, **5y**,...).

Response: We sincerely thank this reviewer for pointing it out. The above problems have been corrected. Please see our revised manuscript.

4. Compared with Scheme 1, the yields of Scheme 2 are generally lower. Can the byproduct with successive B–H insertion be observed? That may be caused by the relatively high activity of donor-acceptor carbenes (similar with 5a to 6 in Scheme 3B).

Response: We deeply thank this reviewer for pointing it out. The byproducts with successive B–H insertion were not observed in our cases. Actually the yields of Scheme 2 are not bad at all, most of them are higher than 80% yields.

5. In the Experimental Section, ^{19}F NMR data should be included for all F-substituted

compounds, and the corresponding spectra should also be included.

Response: We sincerely thank this reviewer for pointing it out. ^{19}F NMR data for all F-substituted compounds have been added to the supporting information. Please see our revised supplementary information.

Reviewer #2 (Remarks to the Author):

In this manuscript, the authors describe an enantioselective synthesis of chiral-at-boron compounds via Cu-catalyzed carbene insertion reactions with tetracoordinated borane-pyridine substrates by using a desymmetrization strategy. This reviewer agrees that although the enantioselective carbene insertion to B–H bonds has recently been studied extensively, the synthesis of chiral-at-boron compounds via this method remained elusive. The reaction design of this work is quite smart in that the utilization of rigid tetracoordinated borane-pyridines as substrates should enhance the chiral discrimination during the key B–H insertion step. Thus, this reviewer recognizes the novelty of this manuscript for published in Nat. Commun. However, a few minor revisions should be addressed before the final acceptance.

Response: We sincerely thank this reviewer for the favorable comments on our work, we really appreciate it.

1. The authors emphasized the importance of the C7-substitution on the borane-pyridine substrates in asymmetric induction. A plausible model was also proposed by performing DFT calculations using both C7-substituted and C7-unsubstituted substrates. However, it should be noted that thiophene- and benzothiophene-derived substrates also participated in the reaction, leading to good enantioselectivity (**5n**, **5o**, 93% ee). The structures of these substrates are quite different from that of the model substrate. Thus, more comments on this phenomenon should be expected in the revised manuscript.

Response: We thank this reviewer for the suggestion. In fact, the enantioselectivity is good for both substituted and unsubstituted substrates (such as **5b** and **5c**). We calculated the electrostatic potential surfaces for thiophene-derived substrates (**5n**, **5o**) and found that they possess similar electron density distribution with the unsubstituted substrates, implying the explanation of the origin of enantioselectivity also works for **5n** and **5o**.

2. This reviewer does not understand on what basis the authors can draw the absolute configuration of the minor product **5a'** explicitly in Table 3. Although DFT calculations showed the energy order of the transition states leading to the two enantiomers (TSSS-5h-L3 and TSRR-5h-L3, Figure 3), this cannot be used as the sole decisive evidence for the determination of absolute configuration. This reviewer suggests not to draw the absolute configuration of **5a'** if SC-XRD data are not available.

Response: We sincerely thank this reviewer for careful readings and raising this question. Per the request, the above question have been solved by removing the absolute configuration of **5a'**. Please see our revised manuscript.

3. The computational level of theory should be described in the main text and the captions of Figures 2 and 3.

Response: We thank this reviewer for this suggestion and the figure captions have been revised accordingly.

4. Further language polishing is necessary since the text is not fluent and grammatic inaccuracy is often encountered: "...but there are also shadows of boron-stereogenic compounds in chiral catalysts and materials" (Page 2); "Motivation and Construction of Boron-stereogenic Compounds Direct on Boron Atom." (Caption of Figure 1) ...

Response: We thank this reviewer for pointing it out. The entire manuscript has been checked carefully and the revisions have been made accordingly. Please see our revised manuscript.

5. Author contributions: "Q.S. & Z.F. wrote the paper". However, these is not a coauthor whose name can be abbreviated as Z.F.

Response: We sincerely thank this reviewer for pointing it out. We are sorry for our negligence. Incorrect information was entered in the author contributions. The above problem has been corrected. Please see our revised manuscript.

Reviewer #3 (Remarks to the Author):

Yu, Song and co-workers described in this article the "Construction of boron stereogenic compounds via enantioselective Cu-catalyzed desymmetric B-H bond insertion reaction".

This paper describes the construction of compounds with a stereogenic boron atom via a copper-catalyzed enantioselective method, resulting from the insertion of diazo compounds into 2-arylpyridine boranes. The method seems general, and the synthesized derivatives are obtained in good yields and with very good enantioselectivities. The use of prochiral diazo reagents also allows to obtain compounds with contiguous stereogenic boron and carbon atoms with very good enantio- and diastereo-selectivities. This study is completed by experimental and theoretical mechanistic studies on the origin of the selectivity, corroborating the experimental results.

Importantly, this article is quite original. It is the second method known in the literature to obtain a stereogenic boron atom by an enantioselective route. Although the reaction of insertion of diazo compounds into the B-H bond is a reactivity that has been described previously, notably by Curran and others, the use of this method here is quite original and leads to a very high control of the enantioselectivity/diastereoselectivity. The results are more or less well presented, the experimental section is complete and well described, and data as NMR spectral data or HPLC traces are of good quality.

Response: We sincerely thank this reviewer for the favorable comments on our work, we really appreciate it.

However, there are some points that definitely need to be improved.

1- As a major point, the manuscript should be proofread by an English speaker. Overall, the sentences are too long, and sometimes become difficult to read and understand, and the manuscript suffers from this.

Response: We thank this reviewer for raising this problem. The entire manuscript has been polished and proofread by an English speaker. Please see our revised manuscript.

2- The term asymmetric should be removed from many sentences, as it is redundant with good stereodescriptive terms such as enantio- or diastereoselective:

*in the abstract: “The previously elusive asymmetric catalytic construction of boron-stereogenic compounds has been achieved through enantioselective desymmetric B–H bond insertion reaction herein”

*in the manuscript: for example, P4, replace asymmetric by enantioselective or in conclusion (P15) remove asymmetric, etc...

Response: We sincerely thank this reviewer for pointing it out. The above problems have been corrected. Please see our revised manuscript.

3- The authors might need to soften some of their claims concerning:

“Of note, the reported asymmetric catalytic strategy achieved by a CuAAC click chemistry by desymmetrization of diacetylative tetracoordinate boron species, it did not directly generate a boron-stereogenic center on the boron atom”

If the authors refer to He's paper (JACS 2021, 143, 16302), this statement is false. The "click" reaction described by He et al. is indeed enantioselective, the boron atom becomes stereogenic in this step. This statement should be deleted or rewritten as it is awkward. Perhaps they meant that they did not directly substitute one group with another directly on the boron atom to reveal a stereogenic boron center?

Response: We deeply thank this reviewer for pointing it out as well as the interpretations. Yes, we want to emphasize that the enantioselective reaction site on boron atom is elusive so far and will be a challenge if the enantioselective reaction site was direct on boron itself. Please see our revised manuscript.

4- Comments on Figure S1, wrongly stated: “IRC calculations verified that the transition state links reactant and product and after hydride transfer step, the boron cation attack carbon to deliver product without any energy penalty”. It is the carbon atom which is nucleophilic (or the organometallic copper species) and which adds itself to the borenium ion, and not the reverse.

Response: We thank this reviewer for the suggestion. We did not specify the “attack” of boron cation is nucleophilic. In order to avoid misunderstandings, we rephrase the sentence as below:

“IRC calculations verified that the calculated transition state links reactant and product and after hydride transfer step, the formation of B–C bond immediately delivers product without any energy penalty.”

Please see our revised SI.

5- One point needs to be clarified concerning the origin of diastereoselectivity. The sentence “There is a synergistic effect between the two stereogenic centers (B and C atoms) instead of inducing effect originated from C-stereogenic center to consecutive B-stereogenic center by the enantioselective B–H bond insertion reaction” is not clear.

It should be explained and detailed.

Response: We thank this reviewer for raising this question. In order to avoid misunderstandings, we rephrase the statement as “Like the aforementioned symmetric carbene insertion, the origin of the chirality of B-stereogenic center is resulted from the interaction between chiral environment of copper-carbene and the asymmetric steric structure of borane which is not solely induced by the construction of C-stereogenic center.”

6- In the ESI, HPLC traces of the racemic compounds from Table 3 and Scheme 2 show only one pair of enantiomers of the major diastereomer. However, without chiral ligand, the d.r. is 2:1 (Table 3). Where is the trace of the second diastereomer?

If the racemic version of the reaction is diastereoselective, this calls the proposed mechanism into question. This point should be reviewed, and the HPLC chromatograms should be shown in full if this was not the case.

Response: We thank this reviewer for raising this question. As we indicated in Table 3, the two diastereomers from the reactions in Table 3 and Scheme 2 were all isolable, we used the major one obtained in pure form from the reaction mixture as both racemic sample and enantioenriched samples to run HPLC, therefore, you could not see the peaks of the second diastereomer in HPLC chromatogram.

In terms of the diastereoselectivity of racemic reaction (entry 9 in Table 3), although the chiral ligand is absent, compared with indole ring, Cl-substituted benzyl ring still has stronger repulsive interaction with ester group which explains the diastereoselectivity (albeit being quite low, about 2:1), being consistent with the proposed mechanism.

Per the request, the HPLC chromatogram of the crude product (with both **5a** and **5a'** without purifications) are listed below, along with the ¹H NMR of the crude product.

6- It misses diastereomeric ratio in schemes 3 and 4

Response: We thank this reviewer for pointing it out. The diastereomeric ratios have been added to Scheme 3 and Scheme 4. Please see our revised manuscript.

After major revisions, I think that the manuscript has the quality, interest, and novelty to be published in Nature Communications.

Response: We sincerely thank this reviewer for the favorable comments on our work, we really appreciate it.

Reviewer #4 (Remarks to the Author):

Song et al. report an enantioselective copper-catalyzed desymmetric B–H bond insertion reaction of 2-arylpyridine-boranes with diazo compounds to construct boron-stereogenic compounds and the stereocontrol of this transformation was suggested by experiments and DFT calculations. However, this strategy is just a derivation of He's work (*J. Am. Chem. Soc.* **2021**, *143*, 16302). They both employed the same catalyst and similar substrates and ligands. The main highlight of this manuscript is switching to B–H bond insertion from CuAAC reaction but both diazo compounds and azides are relative hazardous. The substrate scope is quite narrow. Relatively high enantioselectivity and diastereoselectivity are limited to substrates substituted on C7 which is similar to the results of He's work too. In the part of synthetic utility demonstration, only double B–H bond insertion, reduction of ester and Suzuki coupling reaction are discussed. The author should make comments on the real usefulness of the product therein formed in detail.

In consideration of these aspects, the referee recommends the manuscript should be published in a more specialized journal.

Response: We sincerely thank this reviewer for the critical comments on our work. However, we feel that the issues raised by the reviewer were based some misunderstanding regarding our work. **Firstly**, when He's work (*J. Am. Chem. Soc.* **2021**, *143*, 16302) was online, our work was almost finished, we are waiting for DFT calculations to elucidate the enantioselectivity. It is unfair to say that our work is “just a derivation of He's work”, our group and He's group are independently doing the two works with two different strategies at almost the same time, it is just a coincidence that the products have similar structures at the first glance (actually they are not). **Secondly**, as we mentioned in our manuscript “Inspired by our previous report, cyclic tetracoordinate borons with an *N*-containing ligand would provide rigid scaffold, which will enhance the feasibility for the construction of enantioenriched boron stereogenic compounds.” we choose the cyclic tetracoordinate boranes as substrate due to our previous work (*Chem. Sci.* **2018**, *9*, 7666-7672.), and it originates the rationale design as we performed DFT calculations before exploration of substrate scope, in order to understand why 7-substituted substrates provide excellent enantioselectivities and diastereoselectivities. **Thirdly**, this reviewer commented: “The main highlight of this manuscript is switching to B–H bond insertion from CuAAC reaction but both diazo compounds and azides are relative hazardous.” Copper carbene insertion into B–H bond and CuAAC represent two different types of

reactions with completely distinct mechanisms. It was not a simple switch, it was based on rational design and we found cyclic tetracoordinate boranes are readily accessible, most remarkably, they are stable tetracoordinate organoboranes bearing two same B-H bonds, which perfectly meet our requirement and could serve as an ideal starting material for our purpose, and no one have explored their reactivities, especially on enantioselective catalytic insertion reactions with famous diazo compounds. As the second reviewer commented on our work: *“This reviewer agrees that although the enantioselective carbene insertion to B–H bonds has recently been studied extensively, the synthesis of chiral-at-born compounds via this method remained elusive. The reaction design of this work is quite smart in that the utilization of rigid tetracoordinated borane-pyridines as substrates should enhance the chiral discrimination during the key B–H insertion step.”* And the third reviewer said: *“...The method seems general, and the synthesized derivatives are obtained in good yields and with very good enantioselectivities...Importantly, this article is quite original. It is the second method known in the literature to obtain a stereogenic boron atom by an enantioselective route. Although the reaction of insertion of diazo compounds into the B-H bond is a reactivity that has been described previously, notably by Curran and others, the use of this method here is quite original and leads to a very high control of the enantioselectivity/diastereoselectivity.”* Meanwhile, diazo compounds are one of the most important intermediates and synthons in organic synthesis, and they have been widely implemented in various transformations, and there are so many organic chemists all around the world working on diazo compounds and numerous beautiful reactions, even the enantioselective ones based on diazo compounds have been developed, and azides are in the same situation. So I disagree with this reviewer by saying “they are relative hazardous”. With correct operations, diazo compounds are not hazardous at all, they are under control and valuable versatile synthons.

Fourthly, in terms of usefulness of the products, we had demonstrated it in our manuscript, and further explorations are still under way in our laboratory. Meanwhile, we do not think it will be an issue if the utilities of some brand-new compounds are unknown or elusive. Most of such cases are due to the lack of efficient methods for the construction of them, once practical and general methods have been developed, versatile functions and usefulness of the compounds will be exploited and found quickly. For fundamental research, we pursue a goal with passion to solve very

challenging problems in an area, that is the most important to us.

We still sincerely thank this reviewer for his/her critical but inspiring comments on our manuscript. Hopefully after these clarifications, the reviewer could support our publication on *Nature Communications*.

REVIEWERS' COMMENTS

Reviewer #1 (Remarks to the Author):

The revised manuscript has addressed all of my concerns; the significant modifications will be the important addition to the publication, so I recommend it to be published as it is.

Reviewer #2 (Remarks to the Author):

The authors have addressed the issues this reviewers has raised in the last round of review. This manuscript can now be published in its current form.

Reviewer #3 (Remarks to the Author):

Dear Editor,

the responses and corrections made by Song and co-workers to the pertinent remarks made by the various referees are quite satisfactory. In my opinion, this study has the quality, interest, and novelty to be publish in Nature Communications. For these reasons, I strongly support the publication of this work in your journal.

However, before publication, some errors must be corrected.

1/ In the sentence "In fact, boron-stereogenic compounds can be found not only in natural products, but also in chiral catalysts and materials" reference 26 and corresponding catalyst must be removed: if the catalyst presented in figure 1B is indeed chiral (axial chirality of the binaphthyl derivative), the boron atom is not stereogenic. This is contradictory to the title of the scheme "Boron-stereogenic compounds in nature product, catalyst and material".

To the best of my knowledge, there is no catalyst containing a stereogenic boron atom configurationally stable with a determined absolute configuration. In the case of CBS catalyst, the active form is the trivalent state of oxaborolidine derivative, and the boron atom becomes stereogenic in the "pre-transition-state" (see attached file for schemes: Protonation of oxazaborolidine with triflic acid and for example, pre-transition-state assembly for enantioselective Diels-Alder reactions of 2-substituted α,β -enals).

2/ In the same scheme, the stereogenic boron atom in Boromycin natural product should be drawn as in reference 25, in its tetrahedral form (2 bonds in the plane and the other 2 in front and behind) to highlight the stereogenic boron center.

Point-by-Point Response

REVIEWERS COMMENTS

Reviewer #1 (Remarks to the Author):

The revised manuscript has addressed all of my concerns; the significant modifications will be the important addition to the publication, so I recommend it to be published as it is.

Response: We sincerely thank this reviewer for the favorable comments on our work, we really appreciate it.

Reviewer #2 (Remarks to the Author):

The authors have addressed the issues this reviewer has raised in the last round of review. This manuscript can now be published in its current form.

Response: We sincerely thank this reviewer for the favorable comments on our work, we really appreciate it.

Reviewer #3 (Remarks to the Author):

the responses and corrections made by Song and co-workers to the pertinent remarks made by the various referees are quite satisfactory. In my opinion, this study has the quality, interest, and novelty to be publish in Nature Communications. For these reasons, I strongly support the publication of this work in your journal.

However, before publication, some errors must be corrected.

1/ In the sentence "In fact, boron-stereogenic compounds can be found not only in natural products, but also in chiral catalysts and materials" reference 26 and corresponding catalyst must be removed: if the catalyst presented in figure 1B is indeed chiral (axial chirality of the binaphthyl derivative), the boron atom is not stereogenic. This is contradictory to the title of the scheme "Boron-stereogenic compounds in nature product, catalyst and material".

To the best of my knowledge, there is no catalyst containing a stereogenic boron atom configurationally stable with a determined absolute configuration. In the case of CBS

catalyst, the active form is the trivalent state of oxaborolidine derivative, and the boron atom becomes stereogenic in the “pre-transition-state” (see attached file for schemes: Protonation of oxazaborolidine with triflic acid and for example, pre-transition-state assembly for enantioselective Diels-Alder reactions of 2-substituted α,β -enals).

Response: We deeply thank this reviewer for careful readings and pointing it out. Sorry for our negligence, these problems have been corrected. Please see our revised manuscript.

2/ In the same scheme, the stereogenic boron atom in Boromycin natural product should be drawn as in reference 25, in its tetrahedral form (2 bonds in the plane and the other 2 in front and behind) to highlight the stereogenic boron center.

Response: We deeply thank this reviewer for careful readings and pointing it out. This problem has been corrected. Please see our revised manuscript.